# Challenges and Opportunities in Offline Reinforcement Learning from Visual Observations

**Cong Lu**[*]                                                                cong.lu@stats.ox.ac.uk
*University of Oxford*

**Philip J. Ball**[*]                                                           ball@robots.ox.ac.uk
*University of Oxford*

**Tim G. J. Rudner**                                                           tim.rudner@cs.ox.ac.uk
*University of Oxford*

**Jack Parker-Holder**                                                         jackph@robots.ox.ac.uk
*University of Oxford*

**Michael A. Osborne**                                                         mosb@robots.ox.ac.uk
*University of Oxford*

**Yee Whye Teh**                                                               y.w.teh@stats.ox.ac.uk
*University of Oxford*

**Reviewed on OpenReview:** *https://openreview.net/forum?id=1QqIfGZOWu*

## Abstract

Offline reinforcement learning has shown great promise in leveraging large pre-collected datasets for policy learning, allowing agents to forgo often-expensive online data collection. However, offline reinforcement learning from *visual observations* with continuous action spaces remains under-explored, with a limited understanding of the key challenges in this complex domain. In this paper, we establish simple baselines for continuous control in the visual domain and introduce a suite of benchmarking tasks for offline reinforcement learning from visual observations designed to better represent the data distributions present in real-world offline RL problems and guided by a set of desiderata for offline RL from visual observations, including robustness to visual distractions and visually identifiable changes in dynamics. Using this suite of benchmarking tasks, we show that simple modifications to two popular vision-based online reinforcement learning algorithms, DreamerV2 and DrQ-v2, suffice to outperform existing offline RL methods and establish competitive baselines for continuous control in the visual domain. We rigorously evaluate these algorithms and perform an empirical evaluation of the differences between state-of-the-art model-based and model-free offline RL methods for continuous control from visual observations. All code and data used in this evaluation are open-sourced to facilitate progress in this domain.

> Open-sourced code and data for the V-D4RL benchmarking suite are available at:
> https://github.com/conglu1997/v-d4rl.

---

[*]Equal contribution. Correspondence to cong.lu@stats.ox.ac.uk and ball@robots.ox.ac.uk.

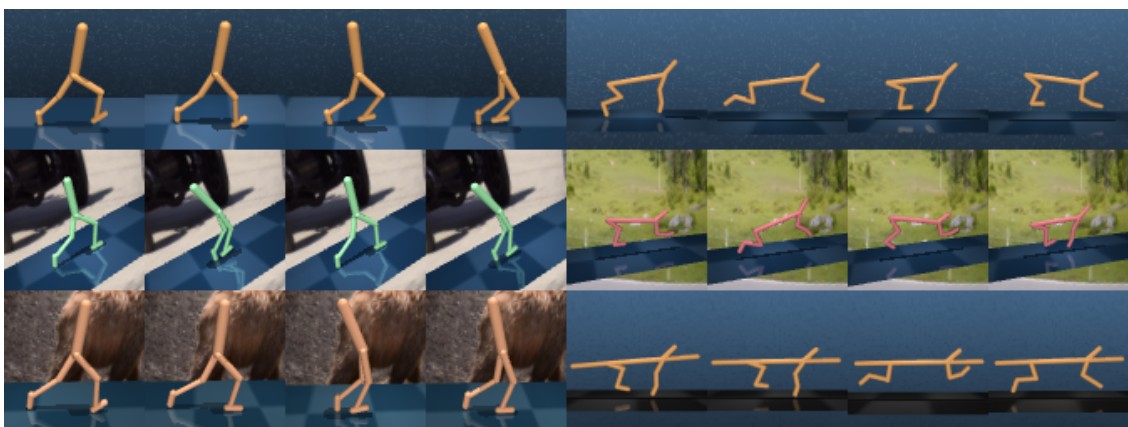

Figure 1: V-D4RL is a benchmarking suite for offline reinforcement learning from visual observations based on the DMControl Suite (Tassa et al., 2020), which includes a comprehensive set of D4RL-style datasets and modalities unique to learning from visual observations.

## 1 Introduction

The reinforcement learning (RL, Sutton & Barto (1992)) paradigm is conventionally characterized as learning from *online interaction* with an environment. However, in many real-world settings, such as robotics or clinical decision-making, online interactions can be expensive or impossible to perform due to physical or safety constraints. *Offline* (or batch) reinforcement learning (Ernst et al., 2005; Levine et al., 2020) aims to address this issue by leveraging pre-collected datasets to train and deploy autonomous agents without requiring online interaction with an environment.

While offline reinforcement learning algorithms, both model-based (Yu et al., 2020b; Kidambi et al., 2020; Argenson & Dulac-Arnold, 2021) and model-free (Kumar et al., 2020; Kostrikov et al., 2021; Fujimoto & Gu, 2021; Wu et al., 2019; Fujimoto et al., 2019; Kumar et al., 2019), have mastered challenging continuous control tasks, most prior works have relied on access to proprioceptive states focusing on standardized benchmarking suites (Fu et al., 2021). In contrast, studies of offline reinforcement learning from *visual observations* (Rafailov et al., 2021; Shah et al., 2021; Chen et al., 2021; Florence et al., 2021) have often focused on disparate tasks with bespoke datasets due to the lack of a well-designed benchmarking suite with carefully evaluated baselines. With this work, we aim to address both of these needs.

Training agents from visual observations offline provides an opportunity to make reinforcement learning more widely applicable to real-world settings, where we have access to vast quantities of visual observations of desirable behaviors. For example, in autonomous driving (Kendall et al., 2018), large quantities of visual offline data already exist but have not been fully utilized (Maddern et al., 2016; Yu et al., 2020a). Similarly, in robotics, data collection is expensive due to costs associated with set-up and human supervision. Effective, transferable offline reinforcement learning could allow us to reuse datasets gathered previously for different tasks or settings for unseen new problems (Chebotar et al., 2021). Unlocking this potential would represent significant progress towards learning general-purpose agents for realistic problems.

To enable the development of effective, robust, and adaptive algorithms for offline RL from visual observations, we present a suite of carefully designed datasets and benchmarking tasks for this burgeoning domain. We use these tasks to establish simple performance baselines, to study how the composition of vision-based offline datasets affects the performance of different types of RL algorithms, and to evaluate the extent to which algorithms for offline RL from visual observations satisfy a set of *desiderata*, including robustness to visual distractions, generalization across environment dynamics, and improved performance at scale. Our evaluation identifies clear failure modes of the baseline methods and highlights opportunities and open problems for future work which can be tackled with our benchmark.

Recent progress in offline reinforcement learning from proprioceptive observations has been driven by well-designed and easy-to-use evaluation testbeds and baselines. We hope that V-D4RL and the analysis in this paper will help facilitate the development of robust RL agents that leverage large, diverse, and often imperfect offline datasets of visual observations across tasks and deployment settings.

The core contributions of this paper are as follows:

1. We present a benchmark for offline RL from visual observations of DMCONTROL SUITE (DMC) tasks (Tassa et al., 2020). This benchmark, **Vision Datasets for Deep Data-Driven RL** (V-D4RL), follows the design principles of the popular D4RL benchmark (Fu et al., 2021), and to our knowledge is the first publicly available benchmark for continuous control which features a wide variety of behavioral policies.

2. We identify **three key desiderata** for realistic offline RL from visual observations: robustness to distractions (Stone et al., 2021), generalization across dynamics (Zhang et al., 2021), and improved performance for offline reinforcement learning at scale. We present a suite of evaluation protocols designed to test whether offline RL algorithms satisfy these desiderata.

3. We establish model-based and model-free baselines for offline RL from visual observations. We do so by modifying two popular online RL algorithms, **DreamerV2** (Hafner et al., 2020b) and **DrQ-v2** (Yarats et al., 2021a), which showcase the relative strengths of model-based and model-free algorithms for offline RL from visual observations. We use these algorithms to provide simple baselines for the aforementioned desiderata to serve as a measure of progress for future advances in this domain.

## 2 Preliminaries

We model the environment as a Markov Decision Process (MDP), defined as a tuple $M = (\mathcal{S}, \mathcal{A}, P, R, \rho_0, \gamma)$, where $\mathcal{S}$ and $\mathcal{A}$ denote the state and action spaces respectively, $P(s'|s, a)$ the transition dynamics, $R(s, a)$ the reward function, $\rho_0$ the initial state distribution, and $\gamma \in (0, 1)$ the discount factor. The standard goal in online reinforcement learning is to optimize a policy $\pi(a|s)$ that maximizes the expected discounted return $\mathbb{E}_{\pi, P, \rho_0} \left[ \sum_{t=0}^{\infty} \gamma^t R(s_t, a_t) \right]$ through interactions with the environment.

In *offline reinforcement learning*, the policy is not deployed in the environment until test time. Instead, the algorithm has access to a fixed dataset $\mathcal{D}_{\text{env}} = \{(s_i, a_i, r_i, s_{i+1})\}_{i=1}^{N}$, collected by one or more behavioral policies $\pi_b$. Following Yu et al. (2020b), we refer to the distribution from which $\mathcal{D}_{\text{env}}$ was sampled as the *behavioral distribution*.

We first describe recent advancements in offline RL and RL from visual observations through the lens of model-based and model-free methods.

### 2.1 Offline Reinforcement Learning Paradigms

**Model-based.** A central problem in offline reinforcement learning is over-estimation of the value function (Sutton & Barto, 1992) due to incomplete data (Kumar et al., 2019). Model-based methods in offline RL provide a natural solution to this problem by penalizing the reward from model rollouts by a suitable measure of uncertainty. Yu et al. (2020b) provide a theoretical justification for this approach by constructing a pessimistic MDP (P-MDP) and lower-bounding the expected true return, $\eta_M(\pi)$, using the error between the estimated and true model dynamics. Since this quantity is usually not available without access to the true environment dynamics, algorithms such as MOPO and MOReL (Yu et al., 2020b; Kidambi et al., 2020) penalize reward with a surrogate measure of uncertainty. These algorithms train an ensemble of $K$ probabilistic dynamics models (Nix & Weigend, 1994) and define a heuristic based on the ensemble predictions. Recent work (Lu et al., 2022) has shown that a better approach to approximating the true dynamics error is to use the standard deviation of the ensemble's mixture distribution instead, as proposed by Lakshminarayanan et al. (2017).

**Model-free.** In the model-free paradigm, we lose the natural measure of uncertainty provided by the model. In lieu of this, algorithms such as CQL (Kumar et al., 2020) attempt to avoid catastrophic overestimation by penalizing actions outside the support of the offline dataset with a wide sampling distribution over the action bounds. Recently, Fujimoto & Gu (2021) have shown that a minimal approach to offline reinforcement learning works in proprioceptive settings, where offline policy learning with TD3 (Fujimoto et al., 2018) can be stabilized by augmenting the loss with a behavioral cloning term.

## 2.2 Reinforcement Learning from Visual Observations

Recent advances in reinforcement learning from visual observations have been driven by use of data augmentation, contrastive learning and learning recurrent models of the environment. We describe the current dominant paradigms in model-based and model-free methods below.

**Model-based (DreamerV2, Hafner et al. (2020b)).** DreamerV2 learns a model of the environment using a Recurrent State Space Model (RSSM, Hafner et al. (2019; 2020a)), and predicts ahead using compact model latent states. The particular instantiation used in DreamerV2 uses model states $s_t$ containing a deterministic component $h_t$, implemented as the recurrent state of a Gated Recurrent Unit (GRU, Chung et al. (2014)), and a stochastic component $z_t$ with a categorical distribution. The actor and critic are trained from imagined trajectories of latent states, starting at encoded states of previously encountered sequences.

**Model-free (DrQ-v2, Yarats et al. (2021b)).** DrQ-v2 is an off-policy algorithm for vision-based continuous control, which uses data-augmentation (Laskin et al., 2020; Yarats et al., 2021c) of the state and next state observations. The base policy optimizer is DDPG (Lillicrap et al., 2016), and the algorithm uses a convolutional neural network (CNN) encoder to learn a low-dimensional feature representation.

# 3 Related Work

There has been significant progress in offline RL, accompanied and facilitated by the creation of several widely used benchmarking suites. We list several here; to our knowledge, no contemporary datasets are both publicly available and feature the same range of behaviors as D4RL, nor do they feature tasks related to distractions or changed dynamics.

**Benchmarks for continuous control on states.** D4RL (Fu et al., 2021) is the most prominent benchmark for continuous control with proprioceptive states. The large variety of data distributions has allowed for comprehensive benchmarking (Kumar et al., 2020; Kidambi et al., 2020; Yu et al., 2020b; 2021; Kostrikov et al., 2021) and understanding the strengths and weaknesses of offline algorithms. Our work aims to establish a similar benchmark for tasks with visual observations. RL Unplugged (Gulcehre et al., 2020) also provides public visual data on DMControl tasks but only provides mixed data which is filtered on the harder locomotion suite. COMBO (Yu et al., 2021) also provides results on the DMC walker-walk environment however does not benchmark over a full set of behavioral policies and the datasets are not publicly available at the time of writing.

**Analysis on characteristics of offline datasets.** Recent work (Florence et al., 2021) has sought to understand when offline RL algorithms outperform behavioral cloning in the proprioceptive setting. Kumar et al. (2021) recommend BC for many settings but showed theoretically that offline RL was preferable in settings combining expert and suboptimal data. This finding is corroborated by our analysis (see Tables 1 and 4).

**Vision-based discrete control datasets.** Whilst there has been a lack of suitable benchmarks for vision-based offline continuous control, vision-based datasets for discrete control have been created for Atari (Agarwal et al., 2020). However, at 50M samples per environment, this benchmarking suite can be prohibitive in terms of its computational hardware requirements (see https://github.com/google-research/batch_rl/issues/10). We believe that V-D4RL's 100,000-sample benchmark represents a more achievable task for practitioners.

**Offline vision-based robotics.** Learning control offline from visual observations is an active area of robotics research closely-related to the problems considered in V-D4RL. Interesting pre-collected datasets exist in this domain but with a primary focus on final performance rather than comprehensive benchmarking on different data modalities; we list some of these here. In QT-Opt (Kalashnikov et al., 2018), data is initially gathered with a scripted exploration policy and then re-collected with progressively more successful fine-tuned robots. Chebotar et al. (2021) follow a similar setup, and explores representation learning for controllers. Mandlekar et al. (2021); Cabi et al. (2020) consider learning from combined sets of human demonstrations, data from scripted policies and random data.

## 4 Baselines for Offline Reinforcement Learning from Visual Observations

In this section, we begin by motivating our creation of a new benchmark (V-D4RL), introduce our new simple baselines combining recent advances in offline RL and vision-based online RL, and present a comparative evaluation of current methods on V-D4RL. Comprehensive and rigorous benchmarks are crucial to progress in nascent fields. To our knowledge, the only prior work that trains vision-based offline RL agents on continuous control tasks is LOMPO (Rafailov et al., 2021). We analyze their datasets in Section 4.3.1 and find they do not conform to standard D4RL convention.

### 4.1 Adopting D4RL Design Principles

In this section, we outline how to generate D4RL-like vision-based datasets for V-D4RL. To generate offline datasets of visual observations, we consider the following three DMCONTROL SUITE (DMC) environments (these environments are easy, medium and hard respectively (Yarats et al., 2021a)):

- **walker-walk**: a planar walker is rewarded for being upright and staying close to a target velocity.
- **cheetah-run**: a planar biped agent is rewarded linearly proportional to its forward velocity.
- **humanoid-walk**: a 21-jointed humanoid is rewarded for staying close to a target velocity. Due to the huge range of motion styles possible, this environment is *extremely challenging* with many local minima and is included as a stretch goal.

From these environments, we follow a D4RL-style procedure in considering five different behavioral policies for gathering the data. As in D4RL, the base policy used to gather the data is Soft Actor–Critic (SAC, Haarnoja et al. (2018)) on the proprioceptive states. We consider the following five settings:

- **random**: Uniform samples from the action space.
- **medium-replay (mixed)**: The initial segment of the replay buffer until the SAC agent reaches medium-level performance.
- **medium**: Rollouts of a fixed medium-performance policy.
- **expert**: Rollouts of a fixed expert-level policy.
- **medium-expert (medexp)**: Concatenation of medium and expert datasets above.

We provide precise specifications for medium, mixed and expert in Section 4.1.1 and discuss the choice of offline behavioral policy in Section 4.1.2. By default, each dataset consists of 100,000 total transitions (often $10\times$ less than in D4RL) in order to respect the memory demands of vision-based tasks. The cheetah and humanoid medium-replay datasets consist of 200,000 and 600,000 transitions respectively due to the increased number of samples required to train policies on these environments. Full statistics of each dataset are given in Appendix A. In Section 5, we further extend the original V-D4RL datasets to study problem settings with multiple tasks and visual distractions as illustrated in Figure 1.

### 4.1.1 Data Generation Policy

To create the offline medium and expert datasets, we first train SAC (Haarnoja et al., 2018) policies on the proprioceptive states until convergence, taking checkpoints every 10,000 frames of interaction. We use a frame skip of 2, the default in other state-of-the-art vision RL algorithms (Hafner et al., 2020b; Yarats et al., 2021a). Medium policies are defined as the first saved agent during training that is able to consistently

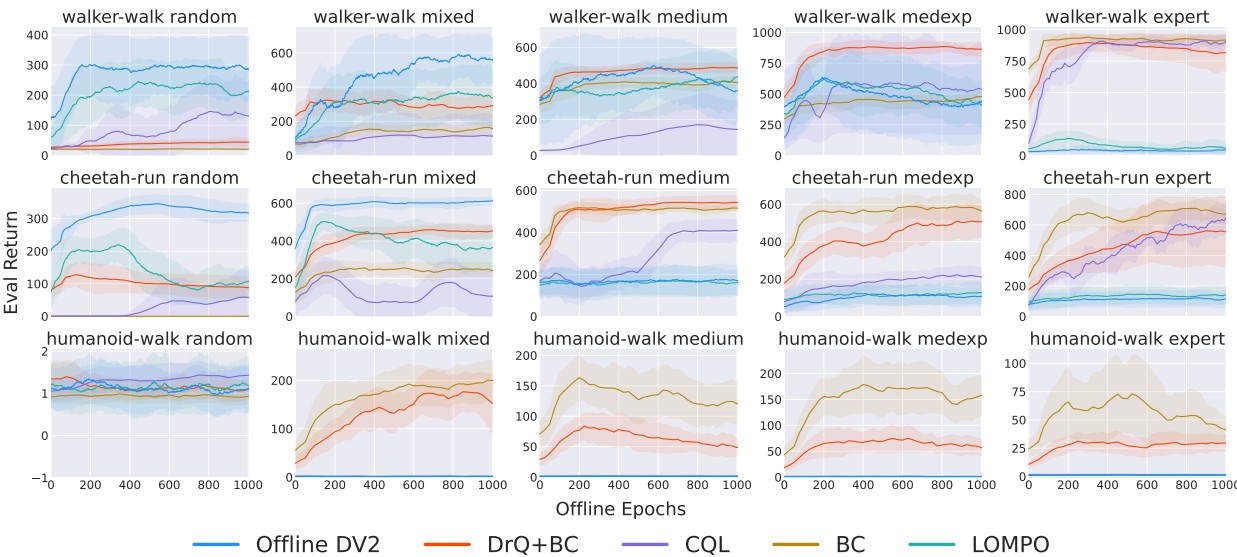

Figure 2: Rigorous comparison on the V-D4RL benchmark, each setting is averaged over six random seeds with error bar showing one standard deviation. Total gradient steps are normalized under epochs, and we plot the un-normalized evaluated return. We note that the model-free and BC baselines are far more stable than the model-based.

achieve above 500 reward in the environment. Expert policies are defined as those that have converged in the limit. For DMControl tasks, this typically means near 1,000 reward on the environment they were trained on. We confirm these thresholds are reasonable, as we observe a noticeable gap between the behavior of the medium and expert-level policies.

In order to generate the offline visual observations, we deploy the proprioceptive agents in the environment, and save the visual observations rendered from the simulator instead of the proprioceptive state. This provides us with the flexibility to *generate observations of any size* without having to retrain for that resolution (e.g., $84 \times 84$ or $64 \times 64$). As was done in D4RL, we generate data using a stochastic actor, which involves sampling actions from the Gaussian conditional distribution of the SAC policy, featuring a parameterized variance head that determines the amount of action stochasticity at each state.

For the mixed datasets, we simply store the replay buffer of the medium agent when it finishes training, and convert the proprioceptive observations into visual observations.

### 4.1.2 Choice of Offline Behavioral Policy

We choose proprioceptive SAC as our behavioral policy $\pi_b$ mirroring Fu et al. (2021); Rafailov et al. (2021). We found that using online DrQ-v2 as the offline behavioral policy made the tasks significantly easier to learn for all agents. This suggests that the proprioceptive agent may learn behavior modes that are less biased towards being easy under vision-based methods; for instance, DrQ-v2 may be biased towards behavior modes that induce fewer visual occlusions compared to proprioceptive SAC.

### 4.2 Baselines

We show that for the two state-of-the-art online vision-based RL algorithms described in Section 2.2, simple adjustments from the proprioceptive literature suffice to transfer them to the offline setting. In Section 4.3, we demonstrate that these baselines provide a new frontier on our benchmark and on prior datasets. Additional details and hyperparameters for our algorithms are given in Appendix B.

**Model-based.** For DreamerV2, we follow Sekar et al. (2020) in constructing a bootstrap ensemble for the dynamics model, which allows us to naturally define a penalty for the reward based on the dynamics

Table 1: Final performance on V-D4RL averaged over six random seeds, with one standard deviation error. Evaluated return is mapped from $[0, 1000]$ to $[0, 100]$. Our model-based method does best on the diverse low-reward datasets, model-free on the diverse high-reward datasets, and behavioral cloning on the narrow expert data. Furthermore, we provide the mean return of the dataset for reference; full statistics are included in Appendix A.

| Environment | | Offline DV2 | DrQ+BC | CQL | BC | LOMPO | Data Mean |
|---|---|---|---|---|---|---|---|
| walker-walk | random | **28.7** ±**13.0** | 5.5 ±0.9 | 14.4 ±12.4 | 2.0 ±0.2 | **21.9** ±**8.1** | 4.2 |
| | mixed | **56.5** ±**18.1** | 28.7 ±6.9 | 11.4 ±12.4 | 16.5 ±4.3 | 34.7 ±19.7 | 14.5 |
| | medium | 34.1 ±19.7 | **46.8** ±**2.3** | 14.8 ±16.1 | 40.9 ±3.1 | 43.4 ±11.1 | 44.0 |
| | medexp | 43.9 ±34.4 | **86.4** ±**5.6** | 56.4 ±38.4 | 47.7 ±3.9 | 39.2 ±19.5 | 70.4 |
| | expert | 4.8 ±0.6 | 68.4 ±7.5 | 89.6 ±6.0 | **91.5** ±**3.9** | 5.3 ±7.7 | 97.0 |
| cheetah-run | random | **31.7** ±**2.7** | 5.8 ±0.6 | 5.9 ±8.4 | 0.0 ±0.0 | 11.4 ±5.1 | 0.7 |
| | mixed | **61.6** ±**1.0** | 44.8 ±3.6 | 10.7 ±12.8 | 25.0 ±3.6 | 36.3 ±13.6 | 19.1 |
| | medium | 17.2 ±3.5 | **53.0** ±**3.0** | 40.9 ±5.1 | 51.6 ±1.4 | 16.4 ±8.3 | 52.4 |
| | medexp | 10.4 ±3.5 | 50.6 ±8.2 | 20.9 ±5.5 | **57.5** ±**6.3** | 11.9 ±1.9 | 70.7 |
| | expert | 10.9 ±3.2 | 34.5 ±8.3 | 61.5 ±4.3 | **67.4** ±**6.8** | 14.0 ±3.8 | 89.1 |
| humanoid-walk | random | 0.1 ±0.0 | 0.1 ±0.0 | 0.2 ±0.1 | 0.1 ±0.0 | 0.1 ±0.0 | 0.1 |
| | mixed | 0.2 ±0.1 | **15.9** ±**3.8** | 0.1 ±0.0 | **18.8** ±**4.2** | 0.2 ±0.0 | 27.6 |
| | medium | 0.2 ±0.1 | 6.2 ±2.4 | 0.1 ±0.0 | **13.5** ±**4.1** | 0.1 ±0.0 | 57.3 |
| | medexp | 0.1 ±0.0 | 7.0 ±2.3 | 0.1 ±0.0 | **17.2** ±**4.7** | 0.2 ±0.0 | 71.6 |
| | expert | 0.2 ±0.1 | 2.7 ±0.9 | 1.6 ±0.5 | **6.1** ±**3.7** | 0.1 ±0.0 | 85.8 |

uncertainty as in Yu et al. (2020b). We adopt an analogous approach to that studied in Lu et al. (2022); Sekar et al. (2020) and use the mean-disagreement of the ensemble. Thus, the reward at each step becomes:

$$\tilde{r}(s,a) = r(s,a) - \lambda \sum_{k=1}^{K} (\mu_\phi^k(s,a) - \mu^*(s,a))^2, \tag{1}$$

where $\lambda$ is a penalty weight and $\mu^*(s,a) = \frac{1}{K}\sum_{k=1}^{K}\mu_\phi^k(s,a)$ is the mean over the dynamics ensemble. Instead of interleaving model-training steps and policy optimization steps, we simply perform one phase of each. We refer to this algorithm as **Offline DV2**.

**Model-free.** For DrQ-v2, we note that the base policy optimizer shares similarities with TD3 (Fujimoto et al., 2018), which has recently been applied effectively in offline settings from proprioceptive states by simply adding a regularizing behavioral-cloning term to the policy loss, resulting in the algorithm TD3+BC (Fujimoto & Gu, 2021). Concretely, the policy objective becomes:

$$\pi = \underset{\pi}{\mathrm{argmax}} \, \mathbb{E}_{(s,a)\sim\mathcal{D}_{\mathrm{env}}} \left[ \lambda Q(s,\pi(s)) - (\pi(s) - a)^2 \right], \tag{2}$$

where $\lambda$ is a normalization term, $Q$ is the learned value function and $\pi$ is the learned policy. We apply the same regularization to DrQ-v2, and call this algorithm: **DrQ+BC**.

**Prior work.** Since DrQ-v2 is an actor-critic algorithm, we may also use it to readily implement the **CQL** (Kumar et al., 2020) algorithm by adding the CQL regularizers to the $Q$-function update. We additionally compare against **LOMPO** (Rafailov et al., 2021), and behavioral cloning (**BC**, Bain & Sammut (1995); Bratko et al. (1995)), where we apply supervised learning to mimic the behavioral policy. Offline DV2 is closely related to LOMPO as both use an RSSM (Hafner et al., 2019; 2020a) as the fundamental model, however, Offline DV2 is based on the newer discrete RSSM with an uncertainty penalty more closely resembling the ensemble penalties in supervised learning (Lakshminarayanan et al., 2017) and uses KL balancing during training (Hafner et al., 2020b).

Table 2: We confirm that our simple baselines outperform LOMPO on the original walker-walk data provided by Rafailov et al. (2021). We report final performance mapped from $[0, 1000]$ to $[0, 100]$ averaged over six random seeds.[1] We show our baselines are more performant than LOMPO on their benchmark. CQL were numbers taken from Rafailov et al. (2021).

| LOMPO Dataset | LOMPO | Offline DV2 | DrQ+BC | CQL |
|---|---|---|---|---|
| medium-replay | 61.3 $\pm$9.1 | **76.3 $\pm$3.1** | 31.1 $\pm$3.7 | 14.7 |
| medium-expert | 69.0 $\pm$24.1 | 72.3 $\pm$20.1 | **73.3 $\pm$3.5** | 45.1 |
| expert | 52.4 $\pm$35.7 | 59.4 $\pm$26.6 | **90.8 $\pm$2.2** | 40.3 |

### 4.3 Comparative Evaluation

We now evaluate the five algorithms described in Section 4.2 on a total of fifteen datasets. To provide a fair evaluation, we provide full training curves for each algorithm in Figure 2 and summarize final performance with error in Table 1. Since no online data collection is required, we measure progress through training via an "offline epochs" metric which we define in Appendix C.

Table 1 shows a clear trend: Offline DV2 is the strongest on the random and mixed datasets, consisting of lower-quality but diverse data, DrQ+BC is the best on datasets with higher-quality but still widely-distributed data and pure BC outperforms on the high-quality narrowly-distributed expert data. We see from Table 1 and Figure 2 that DrQ+BC is extremely stable across seeds and training steps and has the highest overall performance. CQL is also a strong baseline, especially on expert data, but requires significant hyperparameter tuning per dataset, often has high variance across seeds, and is also far slower than DrQ+BC to train. Finally, no algorithm achieves strong performance on the challenging humanoid datasets, mirroring the online RL challenges (Hafner et al., 2020b; Yarats et al., 2021a), with only the supervised BC and, by extension, DrQ+BC showing marginal positive returns.

Perhaps surprisingly, Offline DV2 learns mid-level policies from random data on DMC environments. Furthermore, the random data are more challenging than their D4RL equivalents because there is no early termination, and thus mostly consists of uninformative failed states; this shows the strength of model-based methods in extracting signal from large quantities of suboptimal data. On the other hand, Offline DV2 is considerably weaker on the expert datasets that have narrow data distributions. For these environments, we find the uncertainty penalty is uninformative, as discussed in Appendix D.4.1.

Taking all these findings into consideration leads us to our first open problem, which we believe continued research using our benchmark can help to answer:

> **Challenge 1**: Can a single algorithm outperform both the model-free and model-based baselines, and produce strong performance across *all* offline datasets?

#### 4.3.1 Comparison to the LOMPO Benchmark

For a fair comparison to LOMPO, we also benchmark on the data used in Rafailov et al. (2021) on the DMC Walker-Walk task. In the LOMPO benchmark, the datasets are limited to three types: {medium-replay, medium-expert and expert}. We provide final scores in Table 2. While LOMPO struggles on V-D4RL, it performs reasonably on its own benchmark. However, LOMPO is still outperformed by Offline DV2 on all datasets, whereas DrQ+BC is the best on two datasets.

We may explain the relative strength of LOMPO on this benchmark by noting that the medium-expert dataset used by Rafailov et al. (2021) is described as consisting of the second half of the replay buffer after the agent reaches medium-level performance, thus containing far more diverse data than a bimodal D4RL-style concatenation of two datasets. Furthermore, the expert data is far more widely distributed than that of a standard SAC expert, as we confirm in the statistics in Table 6 of Appendix A.

---

[1]It is unclear how the original scores in Rafailov et al. (2021) were normalized.

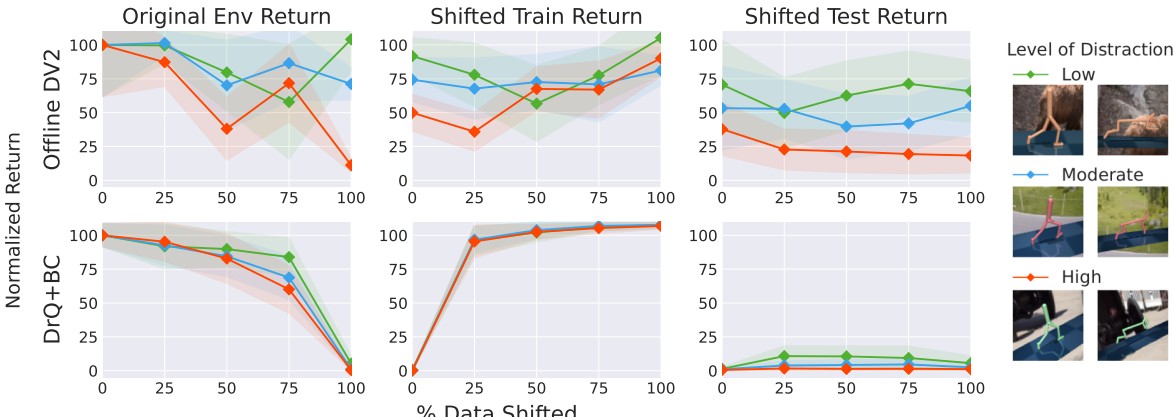

Figure 3: Both DrQ+BC and Offline DV2 readily support training datasets with different distractions (mixture of original and shifted train). Offline DV2 additionally shows the ability to generalize to unseen distractions (shifted test) whereas DrQ+BC is more brittle. Return is normalized against unshifted performance without distractions from Table 1 and averaged over six random seeds. Unnormalized returns are provided in Tables 11 and 12 in Appendix D.1.

## 5 Desiderata for Offline RL from Visual Observations

A key ingredient in recent advances in deep learning is the use of large and diverse datasets, which are particularly prevalent in vision, to train models. Such datasets enable the learning of *general* features that can be successfully transferred to downstream tasks, even when the original task bears little immediate similarity with the transferred task (Xie & Richmond, 2019; Peters et al., 2019). This is a clear rationale for adopting visual observations in offline RL; by leveraging large quantities of diverse high-dimensional inputs, we should be able to learn generalizable features and agents for control. However, combining rich visual datasets with RL presents its own unique challenges. In this section, we present important desiderata that highlight this, and conclude each with an open problem that requires further research.[2]

### 5.1 Desideratum: Robustness to Visual Distractions

One desideratum for offline RL from visual observations is the ability to learn a policy from data collected under multiple different settings. For example, quadrupedal robots deployed at different times of day (e.g., morning and night) will likely gather data having significantly different visual properties, such as brightness and range of visibility. Although the robot may produce similar behaviors in both deployments, superficial differences in visual outputs present a host of opportunities for the agent to learn spurious correlations that prevent generalization (Song et al., 2020; Raileanu & Fergus, 2021).

A key opportunity that arises is the potential to *disentangle* sources of distractions through training on multiple settings, facilitating the learning of general features. By contrast, proprioceptive observations do not generally have distractions, as they are typically low-dimensional and engineered for the task of interest (Lesort et al., 2018). This also limits their ability to transfer, as it is unclear how to incorporate features learned under one set of agent geometries to another.

To test a simplified version of this challenge, we train our baseline agents using newly created datasets featuring varying visual augmentations from the Distracting Control Suite (Stone et al., 2021). This suite provides three levels of distractions (i.e., low, moderate, high), and each distraction represents a shift in the data distribution. We subsequently refer to the level of distraction as the "shift severity" (Schneider et al., 2020). The offline datasets are then constructed as mixtures of samples from the original environment

---

[2]As CQL is quite sensitive to hyperparameters per environment, in the following sections we use the more robust Offline DV2 and DrQ+BC algorithms.

*without distractions* and samples from an environment with a *fixed distraction* level. The learned policies are then evaluated on test environments featuring unseen distractions of the same shift severity.

To create these datasets, we note that the visual distractions from the Distracting Control Suite do not manifest themselves in the proprioceptive states, so we can naturally generate the shifted visual observations by simply rendering the distractions on top of the existing visual observations. Thus, we can simply use the same saved checkpoints as in Section 4.1.1 for the standard datasets.

We compare the baseline algorithms, Offline DV2 and DrQ+BC on datasets that they excel on in Section 4 and Section 5.3: walker-walk random with 100,000 datapoints and cheetah-run medium-expert with 1 million respectively. We visualize returns normalized by this unshifted performance in Figure 3 to show the generalization of each algorithm. We further present full tabular results for Offline DV2 and DrQ+BC in Table 11 and Table 12 respectively in Appendix D.1. The highlighted base unshifted results in both tables are the same as in Table 1 for Offline DV2 and Table 4 for DrQ+BC.

Offline DV2 is able to accommodate datasets with mixed distractions and generalizes reasonably well to unseen test distractions, especially when trained with 'low' and 'moderate' levels of shift severity. Similarly, DrQ+BC is robust to multiple training distractions, with little to no degradation in performance on mixed datasets. However, in contrast, the policy learned is brittle with respect to unseen distractions, and performs significantly worse on the test environments.

Overall, both Offline DV2 and DrQ+BC represent strong baselines for mixed datasets. Interestingly, Offline DV2 demonstrates strong generalization to unseen distractions. This can be explained by generalization that occurs in the trained world-model, which uses a self-supervised loss; we discuss reasons behind this in Appendix D.5. This could be improved even further with recent reconstruction-free variants of DreamerV2 (Okada & Taniguchi, 2021; Nguyen et al., 2021) which have shown robustness to distractions. On the other hand, we observe DrQ+BC generalizes poorly to unseen distractions, presenting a direction for future work using our datasets to learn robust *model-free* agents. This directly leads us to our next open problem:

> **Challenge 2**: How can we improve the generalization performance of offline model-free methods to unseen visual distractions?

### 5.2 Desideratum: Generalization Across Environment Dynamics

Another desideratum of offline RL from visual observations is learning policies that can generalize across multiple dynamics. This challenge manifests in three clear ways. Firstly, we will likely collect data from multiple agents that each have different dynamics, and must therefore learn a policy that can perform well when deployed on any robot that gathered the data (i.e., train time robustness). Secondly, we may be provided with asymmetric data, featuring scarce coverage of particular dynamics, and therefore require the

Table 3: Evaluation on the DMC-Multitask benchmark using *random* data for Offline DV2 and *medexp* data for DrQ+BC and BC. Normalized performance from [0, 1000] to [0, 100] over six random seeds. Our algorithms learn multitask policies from visual observations, with a slight generalization gap for extrapolation tasks. Different dataset types are used for each algorithm to reflect realistic use cases.

| Algorithm | Environment | | Eval. Return | | |
|---|---|---|---|---|---|
| | | | Train Tasks | Test Interp. | Test Extrap. |
| DrQ+BC | walker | | 90.8 | 91.4 | 65.1 |
| | cheetah | medexp | 71.6 | 65.1 | 43.2 |
| BC | walker | | 61.2 | 61.4 | 47.2 |
| | cheetah | | 69.7 | 61.3 | 39.6 |
| Offline DV2 | walker | random | 24.4 | 25.3 | 24.9 |
| | cheetah | | 31.6 | 31.1 | 31.1 |

Table 4: The reinforcement learning algorithms readily scale to higher dataset sizes, compared to supervised behavioral cloning, with a particular benefit to the *medexp* and *expert* datasets for DrQ+BC and the *random* and *medium* datasets for Offline DV2. Results are averaged over six random seeds, with one standard deviation given as error. The evaluated return is mapped from $[0, 1000]$ to $[0, 100]$, and the fixed-size *mixed* dataset is excluded.

| Environment | | Offline DV2 | | DrQ+BC | | BC | |
|---|---|---|---|---|---|---|---|
| | | 100K | 500K | 100K | 500K | 100K | 500K |
| walker | random | 28.7 ±13.0 | 49.9 ±1.6 | 5.5 ±0.9 | 3.5 ±0.6 | 2.0 ±0.2 | 2.1 ±0.3 |
| | medium | 34.1 ±19.7 | 61.3 ±10.9 | 46.8 ±2.3 | 51.0 ±1.1 | 40.9 ±3.1 | 40.9 ±3.0 |
| | medexp | 43.9 ±34.4 | 38.9 ±28.1 | 86.4 ±5.6 | 94.1 ±2.0 | 47.7 ±3.9 | 48.8 ±5.3 |
| | expert | 4.8 ±0.6 | 7.1 ±5.3 | 68.4 ±7.5 | 94.2 ±2.3 | 91.5 ±3.9 | 95.1 ±2.5 |
| cheetah | random | 31.7 ±2.7 | 40.8 ±4.2 | 5.8 ±0.6 | 10.6 ±0.7 | 0.0 ±0.0 | 0.0 ±0.0 |
| | medium | 17.2 ±3.5 | 39.2 ±14.4 | 53.0 ±3.0 | 57.3 ±1.2 | 51.6 ±1.4 | 52.9 ±1.3 |
| | medexp | 10.4 ±3.5 | 9.7 ±5.0 | 50.6 ±8.2 | 79.1 ±5.6 | 57.5 ±6.3 | 69.6 ±10.6 |
| | expert | 10.9 ±3.2 | 11.3 ±4.7 | 34.5 ±8.3 | 75.3 ±7.5 | 67.4 ±6.8 | 87.8 ±1.9 |
| Average Overall | | 22.7 ±10.1 | 32.3 ±9.3 | 43.9 ±4.6 | 58.1 ±2.6 | 44.8 ±3.2 | 49.7 ±3.1 |
| Percentage Gain | | +42.1% | | +32.5% | | +10.8% | |

ability to leverage data from more abundant sources (i.e., transferability). Thirdly, we may be presented with *unseen* dynamics at deployment time, and must therefore learn a policy that is robust to these changes (i.e., test time robustness).

A key opportunity that arises in visual observations is the improved richness of the underlying dataset compared to proprioceptive data. For instance, *some dynamics changes may be visually obvious* (e.g., changed limb sizes, broken actuators), whereas in the proprioceptive setting, such information may not be available. Without this information, we must turn to meta-RL (Rakelly et al., 2019; Zintgraf et al., 2021) or HiP-MDP (Killian et al., 2017; Zhang et al., 2021) approaches that try to infer the missing information from gathered trajectories, adding complexity to the RL process. In contrast, this information can be contained explicitly in visual observations and should allow adaption to a range of downstream tasks without complex inference methods.

To test this hypothesis, we consider two settings from the MTEnv benchmark (Sodhani et al., 2021) which adapts DMC: cheetah-run with modified torso length and walker-walk with modified leg length. We follow an analogous approach to Zhang et al. (2021) where we consider eight different settings {A - H} ordered in terms of limb length, and construct new offline datasets using {B, C, F, G} in equal proportions as our training data. The settings {A, H} are considered the *extrapolation* generalization environments and {D, E} *interpolation* generalization. Since these environments are different from the original, we retrain SAC policies on the proprioceptive states from the modified tasks and define medium and expert policies in the same way as described in Section 4.1.1.

As before, to provide a comparison on realistic use cases of each algorithm, we evaluate Offline DV2 on random datasets of size 100,000 and DrQ+BC on medium-expert datasets containing one million samples and show the results in Table 3. We see that DrQ+BC learns policies that are suitable for transfer across multiple tasks in both walker and cheetah, and maintains that performance on the interpolation test environments. For the extrapolation environments, we see an average drop of around 30%. While this may represent adequate performance, especially compared to a medium policy, it is a striking drop when compared to performance on in-distribution dynamics. This suggests there is a dynamics generalization gap that remains for model-free methods when extrapolating, and represents clear opportunities for further research.

Offline DV2 displays similar trends (results on medexp datasets are in Appendix D.2). On the random data, Offline DV2 learns a similar quality policy to that on the base environment, but experiences no deterioration in performance on the test environments in walker or cheetah. Thus, we demonstrate the sufficiency of

both Offline DV2 and DrQ+BC as baselines in multitask offline RL *without any modification*; model-based approaches further admit opportunities for zero-shot generalization (Ball et al., 2021).

We now contrast our work to that of Zhang et al. (2021), where a multitask policy was trained using a total of 3.2 million time steps of online data collection. Whilst it is hard to compare offline and online results, our DrQ+BC algorithm uses less data, with 1 million total time steps, and obtains similar extrapolation return on the walker environments. This supports a similar conclusion reached by Kurin et al. (2022) who show that our approach, simply minimizing the sum of the task losses, is drastically underestimated in the literature. As noted before, we suffer a comparatively larger drop in performance, lending further evidence that closing this generalization gap should be prioritized.

In conclusion, we believe there are many further avenues for future research using these benchmarks; an immediate open problem we have identified is as follows:

> **Challenge 3**: How can we improve generalization to new dynamics that are not contained in the offline dataset?

### 5.3 Desideratum: Improved Performance with Scale

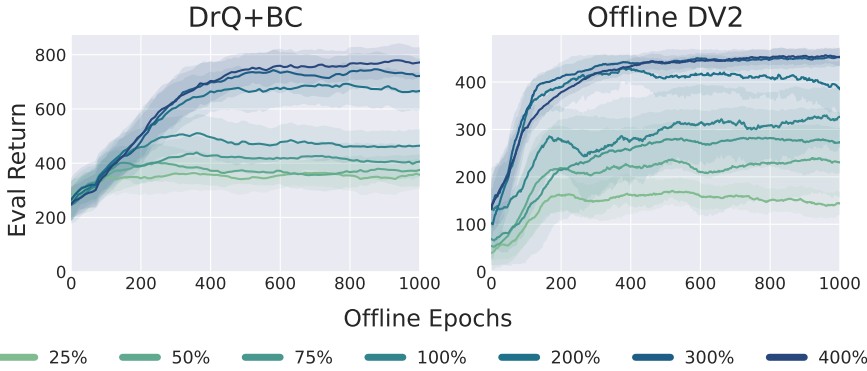

Figure 4: Sensitivity analysis on dataset size for both Offline DV2 (walker-walk random) and DrQ+BC (cheetah-run medexp). Both methods scale with more data, but receive diminishing returns past $2\times$ the original size. Performance averaged over four random seeds.

Learning from large datasets presents huge opportunities for learning general agents for control. To make use of them, we need to understand how our baselines scale with dataset size. We analyze our base choice of 100,000 observations for V-D4RL in Figure 4, where we vary the size of the walker-walk random dataset for Offline DV2 and the cheetah-run medexp dataset for DrQ+BC in the range of $\{0.25\times, \ldots, 4\times\}$ the size of the original dataset. We observe a monotonic increase in the performance of both Offline DV2 and DrQ+BC with increasing dataset size but hit diminishing returns past $2\times$ the original size. We note, perhaps surprisingly, that Offline DV2 can reach $\approx 500$ total return from random data that average $10\times$ less.

For the walker and cheetah datasets with fixed distributions—random, medium, medium-expert and expert— Table 4 shows an average increase of 42.1% for Offline DV2 and 32.5% for DrQ+BC compared to 10.8% for BC when we scale the dataset size to 500,000, showing that the reinforcement learning algorithms make far better use of increased data than supervised behavioral cloning. However, a crucial difference between Offline DV2 and DrQ+BC is that DrQ+BC handles larger offline datasets far more readily. DrQ+BC policy training for the same number of epochs on 500,000 and 100,000 observations takes 8 and 1.6 hours respectively on a V100 GPU. This is significantly quicker than Offline DV2, which takes 10 hours to train on 100,000 observations; we discuss this further in Appendix D.4.1. This significant computational discrepancy leads to a clear open problem:

> **Challenge 4**: How can we scale model-based methods to larger datasets?

## 6 Conclusion and Limitations

In this paper, we took the first steps towards establishing fully open-sourced benchmarking tasks featuring a broad range of behavioral policies and competitive baselines for offline reinforcement learning from visual observations. Until now, work in this space has been nascent, with ad-hoc analyses leading to unclear comparisons. To address the lack of meaningful evaluations and comparative analyses in this space, we provided a set of straightforward and standardized benchmarking tasks that follow popular low-dimensional equivalent experiment setups and presented competitive model-based and model-free baselines. We analyzed key factors that help explain the performance of these approaches, while also demonstrating their ability to generalize in more challenging settings that are unique to visual observations.

With a particular focus on the DeepMind Control Suite, which features a wide array of tasks of varying difficulties, we hope that V-D4RL will be useful to practitioners and researchers alike and that it will provide a springboard for developing offline reinforcement learning methods for real-world continuous-control problems and spark further progress in this space. Complementing and extending V-D4RL to more domains that feature complex manipulation tasks and physical robotics systems presents an exciting direction for future research.

## Acknowledgements

Cong Lu and Tim G. J. Rudner are funded by the Engineering and Physical Sciences Research Council (EPSRC). Philip J. Ball is funded through the Willowgrove Studentship. Tim G. J. Rudner is also funded by a Qualcomm Innovation Fellowship. We are grateful to Rafael Rafailov for sharing the data used with LOMPO and Edoardo Cetin for his NumPy Array replay buffer implementation. We thank Samuel Daulton, Clare Lyle, and Muhammad Faaiz Taufiq for detailed feedback on an early draft of this paper. The authors would also like to thank the anonymous TMLR reviewers for constructive feedback that helped improve the paper.

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

# Supplementary Material

## Table of Contents

## A    Offline Dataset Characteristics

We provide explicit statistics on the returns of each episode for the datasets used in our main evaluation. This provides a reasonable proxy to how diverse each dataset is.

Table 5: Full summary statistics of per-episode return in the V-D4RL benchmark.

|          | Dataset | Timesteps | Mean  | Std. Dev. | Min.  | P25   | Median | P75   | Max.  |
|----------|---------|-----------|-------|-----------|-------|-------|--------|-------|-------|
|          | random  | 100K      | 42.3  | 8.7       | 30.0  | 34.9  | 41.3   | 46.8  | 74.6  |
|          | mixed   | 100K      | 144.5 | 155.9     | 10.9  | 44.3  | 69.4   | 162.4 | 604.9 |
| walker   | medium  | 100K      | 439.6 | 48.4      | 176.2 | 423.1 | 445.5  | 466.7 | 538.0 |
|          | medexp  | 200K      | 704.1 | 267.7     | 176.2 | 445.5 | 538.0  | 969.1 | 990.6 |
|          | expert  | 100K      | 969.8 | 12.4      | 909.2 | 963.9 | 969.1  | 979.5 | 990.6 |
|          | random  | 100K      | 6.6   | 2.6       | 1.1   | 4.7   | 6.3    | 8.4   | 16.3  |
|          | mixed   | 200K      | 191.2 | 144.6     | 2.5   | 48.3  | 191.9  | 303.4 | 473.8 |
| cheetah  | medium  | 100K      | 523.8 | 25.5      | 325.3 | 509.1 | 524.2  | 538.3 | 578.3 |
|          | medexp  | 200K      | 707.0 | 184.9     | 325.3 | 524.2 | 578.3  | 894.1 | 905.7 |
|          | expert  | 100K      | 891.1 | 11.2      | 843.0 | 886.9 | 894.2  | 898.5 | 905.7 |
|          | random  | 100K      | 1.1   | 0.8       | 0.0   | 0.5   | 1.0    | 1.5   | 5.7   |
|          | mixed   | 600K      | 275.7 | 176.1     | 0.0   | 93.7  | 341.7  | 423.1 | 529.0 |
| humanoid | medium  | 100K      | 573.0 | 16.7      | 526.5 | 560.6 | 572.9  | 584.9 | 609.4 |
|          | medexp  | 200K      | 715.9 | 146.1     | 526.5 | 572.9 | 620.6  | 877.4 | 889.8 |
|          | expert  | 100K      | 858.1 | 42.4      | 631.8 | 846.4 | 877.6  | 885.5 | 889.8 |

We compare this to the LOMPO datasets and find that they are more widely distributed, due to the differing data-collection method.

Table 6: Summary statistics of per-episode return in the LOMPO DMC walker-walk datasets.

|        | Dataset | Timesteps | Mean  | Std. Dev. | Min.  | P25   | Median | P75   | Max.  |
|--------|---------|-----------|-------|-----------|-------|-------|--------|-------|-------|
|        | mixed   | 100K      | 208.9 | 144.1     | 33.3  | 75.1  | 172.4  | 340.5 | 496.7 |
| walker | medexp  | 100K      | 674.4 | 92.9      | 501.3 | 596.8 | 679.8  | 752.5 | 869.1 |
|        | expert  | 100K      | 920.6 | 81.6      | 17.9  | 905.9 | 950.3  | 957.9 | 987.0 |

As we see in Table 6 and Figure 5, the LOMPO walker-walk expert dataset has a standard deviation roughly 8x higher than our expert dataset and has an extremely wide [min, max] range. Furthermore, whilst our medexp dataset is bimodal, the LOMPO medexp dataset's returns are a continuous progression. This reflects that the LOMPO data is sampled from the second half of a replay buffer after medium-level performance is attained, akin to the medium-replay (mixed) datasets.

### A.0.1    Broader Issues with Visual Data

Large datasets consisting of images often contain systematic biases, which can damage generalization. The datasets constructed in this paper are all synthetic from simulated reinforcement learning environments. However, as we move towards applying offline RL from visual observations to real-world tasks, it is important to take these potential dangers into account and extend existing work in algorithmic fairness from computer vision to our setting.

## B    Algorithmic Details

We provide additional details for both algorithms here and indicate where our modifications have been made.

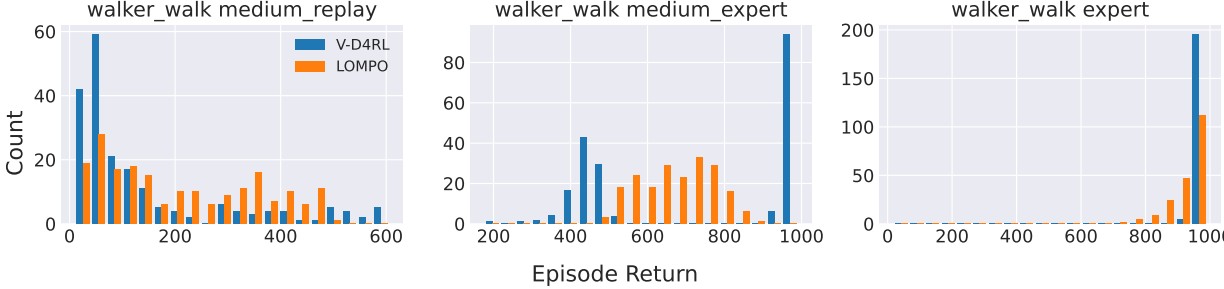

Figure 5: Comparison of the episodic returns from the LOMPO and V-D4RL. We see LOMPO has significantly more diversity in the medium-expert and expert datasets. Note that there is a single episode in the LOMPO expert dataset which has a low return of 17.9.

## B.1 Offline DV2

In the offline setting, it suffices to simply perform one phase of model training and one phase of policy training for the DreamerV2 (Hafner et al., 2020b) algorithm. Each episode in the offline dataset is ordered sequentially to facilitate sequence learning. To this instantiation of DreamerV2, we simply add a reward penalty corresponding to the mean disagreement of the dynamics ensemble. During standard DreamerV2 policy training, imagined latent trajectories $\{(s_\tau, a_\tau)\}_{\tau=t}^{t+H}$ are assigned reward $r_\tau = \mathrm{E}\left[q_\theta\left(\cdot \mid s_\tau\right)\right]$ according to the mean output of the reward predictor. The imagined latent states $s_t$ consist of a deterministic component $h_t$, implemented as the recurrent state of a GRU, and a stochastic component $z_t$ with categorical distribution. The logits of the categorical distribution are computed from an ensemble (with input $h_t$) over which we compute the mean disagreement.

## B.2 DrQ+BC

Here, we simply modify the policy loss term in DrQ-v2 (Yarats et al., 2021a) to match the loss given in Fujimoto & Gu (2021). Following the notation from Yarats et al. (2021a), the DrQ-v2 actor $\pi_\phi$ is trained with the following loss:

$$\mathcal{L}_\phi(\mathcal{D}) = -\mathbb{E}_{\boldsymbol{s}_t \sim \mathcal{D}}\left[Q_\theta\left(\boldsymbol{h}_t, \boldsymbol{a}_t\right)\right]$$

where $\boldsymbol{h}_t = f_\xi\left(\mathrm{aug}\left(\boldsymbol{s}_t\right)\right)$ is the encoded augmented visual observation, $\boldsymbol{a}_t = \pi_\phi\left(\boldsymbol{h}_t\right) + \epsilon$ is the action with clipped noise to smooth the targets $\epsilon \sim \mathrm{clip}\left(\mathcal{N}\left(0, \sigma^2\right), -c, c\right)$. Note that we also do not update encoder weights with the policy gradient. We also train a pair of two fully connected Q-networks, which both use features from a single encoder, and take their minimum when calculating target values and actor losses. The resultant algorithm can be viewed as TD3 (Fujimoto et al., 2018), but with decaying smoothing noise parameters $c$.

In DrQ+BC, this loss becomes:

$$\mathcal{L}_\phi(\mathcal{D}) = -\mathbb{E}_{\boldsymbol{s}_t, \boldsymbol{a}_t \sim \mathcal{D}}\left[\lambda Q_\theta\left(\boldsymbol{h}_t, \boldsymbol{a}_t\right) - \left(\pi_\phi\left(\boldsymbol{h}_t\right) - \boldsymbol{a}_t\right)^2\right]$$

where $\lambda = \frac{\alpha}{\frac{1}{N}\sum_{(h_i, a_i)}|Q(h_i, a_i)|}$ is an adaptive normalization term computed over minibatches. $\alpha$ is a behavioral cloning weight, always set to 2.5 in Fujimoto & Gu (2021), which we also adopt. We experimented performing an extensive grid search over $\alpha$, but did not observe any noticeable benefit in our offline datasets by deviating away from the default value.

## B.3 CQL

Our CQL implementation was also built on top of the DrQv2 codebase for comparability. We use the CQL($\mathcal{H}$) objective with fixed weight, as we found this to be the most performant. This corresponds to choosing the KL-divergence to a uniform prior as the regularizer $\mathcal{R}(\mu)$ in Kumar et al. (2020). Concretely,

the Q-function objective becomes

$$\min_Q \alpha_{\text{CQL}} \mathbb{E}_{\mathbf{s} \sim \mathcal{D}} \left[ \log \sum_{\mathbf{a}} \exp(Q(\mathbf{s}, \mathbf{a})) - \mathbb{E}_{\mathbf{a} \sim \hat{\pi}_\beta(\mathbf{a}|\mathbf{s})}[Q(\mathbf{s}, \mathbf{a})] \right] + \frac{1}{2} \mathbb{E}_{\mathbf{s}, \mathbf{a}, \mathbf{s}' \sim \mathcal{D}} \left[ \left( Q - \hat{\mathcal{B}}^{\pi_k} \hat{Q}^k \right)^2 \right]$$

where $\alpha_{\text{CQL}}$ is a trade-off factor, $\hat{\pi}_\beta$ refers to the empirical behavioral policy and $\hat{\mathcal{B}}^{\pi_k}$ to the empirical Bellman operator that backs up a single sample. This is approximated by taking gradient steps and sampling actions from the given bounds.

## C   Hyperparameter and Experiment Setup

### C.1   Offline DV2

Our Offline DV2 implementation was built on top of the official DreamerV2 repository at: `https://github.com/danijar/dreamerv2` with minor modifications. The code was released under the MIT License. Table 7 lists the hyperparameters used for Offline DV2. For other hyperparameter values, we used the default values in the DreamerV2 repository.

Table 7: Offline DV2 hyperparameters.

| Parameter | Value(s) |
|---|---|
| ensemble member count (K) | 7 |
| imagination horizon (H) | 5 |
| batch size | 64 |
| sequence length (L) | 50 |
| action repeat | 2 |
| observation size | [64, 64] |
| discount ($\gamma$) | 0.99 |
| optimizer | Adam |
| learning rate | {model $= 3 \times 10^{-4}$, actor-critic $= 8 \times 10^{-5}$} |
| model training epochs | 800 |
| agent training epochs | 2,400 |
| uncertainty penalty | mean disagreement |
| uncertainty weight ($\lambda$) | in [3, 10] |

We found a default value of $\lambda = 10$ works for most settings. The only settings where this changes are $\lambda = 3$ for both random datasets and $\lambda = 8$ for the walker-walk mixed dataset.

For the penalty choice, we chose mean disagreement of the ensemble because it comprises one half of the ensemble variance, which was shown to be an optimal choice for offline model-based reinforcement learning in Lu et al. (2022). We found that the other component of the ensemble variance, the average variance over the ensemble, was uninformative and so discarded it.

#### C.1.1   Understanding the impact of hyperparameters

We run additional experiments to understand whether the key default online hyperparameters perform well in the offline setting; for proprioceptive environments, this is not always the case (Lu et al., 2022). Using V-D4RL we are able to better understand the impact of these design choices, and we show the results in Table 8. Concretely, the optimal hyperparameters are very different between Offline DV2 and the online DreamerV2. In the offline setting, it is better to increase in batch size from 16 to 64, and a decrease the imagined horizon (H) from 15 to 5. With our setup, the increase in batch size takes around 76% more wall-clock time per batch, but this leads to a huge gain in performance when normalizing for the actual amount of data that is trained on. We believe the reduction in horizon is required due to the lack of online 'remedial' sampling (i.e., sampling data from the true environment that corrects for errors in the imagined rollouts),

thus resulting in exploitation when training solely on the offline data, even with an uncertainty penalty. However, as was also observed in Lu et al. (2022), choosing too low a horizon (i.e., H = 2) is ill-advised, as this results in over-reliance on the offline state distribution, with less chance of policy improvement. Consequently, we choose batch_size = 16 and H = 5 for all our experiments.

Table 8: Ablations on hyperparameters of Offline DV2 where they differ from the online DreamerV2. We report final performance mapped from $[0, 1000]$ to $[0, 100]$ averaged over six seeds.

| Environment | | Default | batch_size=16 | H=2 | H=15 |
|---|---|---|---|---|---|
| | random | 28.7 ±13.0 | 22.6 ± 8.4 | 31.9 ± 7.6 | 16.9 ± 8.5 |
| | mixed | 56.5 ±18.1 | 27.4 ±16.5 | 54.7 ±14.5 | 45.5 ±15.1 |
| walker | medium | 34.1 ±19.7 | 33.3 ±21.4 | 47.8 ±18.8 | 21.0 ±20.3 |
| | medexp | 43.9 ±34.4 | 24.5 ±30.1 | 22.4 ±18.1 | 22.9 ±25.7 |
| | expert | 4.8 ± 0.6 | 2.7 ± 1.0 | 3.4 ± 0.6 | 3.1 ± 0.8 |
| | random | 31.7 ± 2.7 | 29.3 ± 4.0 | 29.8 ± 4.3 | 32.4 ± 3.7 |
| | mixed | 61.6 ± 1.0 | 53.7 ± 6.9 | 56.5 ± 2.7 | 61.3 ± 2.8 |
| cheetah | medium | 17.2 ± 3.5 | 17.4 ± 8.0 | 17.2 ± 6.2 | 14.3 ± 4.7 |
| | medexp | 10.4 ± 3.5 | 9.4 ± 5.3 | 8.1 ± 3.8 | 8.7 ± 5.2 |
| | expert | 10.9 ± 3.2 | 11.1 ± 5.2 | 8.1 ± 4.8 | 9.7 ± 3.7 |
| Average Change | | - | -21.6% | -6.1% | -15.8% |

## C.2   DrQ+BC

Our DrQ+BC implementation was built on top of the official DrQ-v2 repository at: `https://github.com/facebookresearch/drqv2`. The code was released under the MIT License. Table 9 lists the hyperparameters used for DrQ+BC. For other hyperparameter values, we used the default values in the DrQ-v2 repository. Due to the size of some of our offline datasets, we found the default replay buffer would not scale to the offline datasets. Thus, we used a NumPy (Harris et al., 2020) array implementation instead.

Table 9: DrQ+BC hyperparameters.

| Parameter | Value |
|---|---|
| batch size | 256 |
| action repeat | 2 |
| observation size | [84, 84] |
| discount ($\gamma$) | 0.99 |
| optimizer | Adam |
| learning rate | $1 \times 10^{-4}$ |
| agent training epochs | 256 |
| $n$-step returns. | 3 |
| Exploration stddev. clip | 0.3 |
| Exploration stddev. schedule. | linear(1.0, 0.1, 500000) |
| BC Weight ($\alpha$) | 2.5 |

We also further tuned $\alpha$ within {1.5, 2.5, 3.5} but as Fujimoto & Gu (2021) found, we did not observe any noticeable benefit from deviating away from the default value for $\alpha$.

## C.3   Behavioral Cloning

Our BC implementation shares the exact same policy network and hyperparameters in DrQ+BC but just minimizes MSE on the offline data. Consequently, we must also optimize the learned encoder using the supervised learning loss (unlike in DrQ+BC, where the TD-loss only contributes to the encoder representation learning, and not the policy loss).

## C.4   CQL

Similarly to BC, our CQL implementation is also based on the same networks and hyperparameters in DrQ+BC. CQL introduces one extra hyperparameter, the trade-off factor $\alpha_{\text{CQL}}$. We perform a hyperparameter sweep for this over the range: $\{0.5, 1, 2, 5, 10, 20\}$. We chose the following values per environment:

Table 10: CQL trade-off factor per environment for Walker and Cheetah. Humanoid omitted as all choices performed equally.

| Dataset | | Trade-off Factor ($\alpha_{\text{CQL}}$) |
|---|---|---|
| | random | 0.5 |
| | mixed | 0.5 |
| walker | medium | 2 |
| | medexp | 2 |
| | expert | 5 |
| | random | 0.5 |
| | mixed | 0.5 |
| cheetah | medium | 10 |
| | medexp | 1 |
| | expert | 20 |

## C.5   LOMPO

For our LOMPO evaluation, we used the official repository at: `https://github.com/rmrafailov/LOMPO`. The code was open-sourced without license. We perform a hyperparameter search over the uncertainty weight $\lambda$ in the range $\{1, 5\}$. The default value used in Rafailov et al. (2021) is $\lambda = 5$, but we found that $\lambda = 1$ worked better for random datasets. The accompanying data for the DMC Walker-Walk data from Rafailov et al. (2021) was very kindly provided by the authors.

## C.6   Computational Cost

The experiments in this paper were run on NVIDIA V100 GPUs. On the standard v-d4rl 100K datasets, DrQ+BC took 1.6 hours, Offline DV2 took 10 hours, and CQL took 12 hours.

Since we wish to compare several offline algorithms using the same dataset, we define a notion of "offline epoch" for all algorithms to show training performance over time. We simply normalize the total number of gradient steps, so training progress falls within $[0, 1000]$.

# D   Further Results and Ablation Studies

## D.1   Full Tabular Distracted Results

We further present full tabular results from Figure 3 in Section 5.1 for both Offline DV2 and DrQ+BC in Table 11 and Table 12 respectively. The highlighted base unshifted results in both tables are the same as in Table 1 for Offline DV2 and Table 4 for DrQ+BC.

Table 11: Offline DV2 shows surprisingly good generalization to unseen distractions and can handle mixed datasets. Final mean performance is averaged over six random seeds and base undistracted performance is highlighted. The environment used is walker-walk random.

| Shift Severity | % Shifted | Evaluation Return | | |
|---|---|---|---|---|
| | | Original | Distraction Train | Distraction Test |
| low | 0% | 28.7 | 25.9 | 20.0 |
| | 25% | 28.1 | 22.0 | 14.0 |
| | 50% | 22.5 | 16.0 | 17.7 |
| | 75% | 16.3 | 21.9 | 20.1 |
| | 100% | 29.4 | 29.7 | 18.6 |
| moderate | 0% | 28.7 | 21.0 | 15.1 |
| | 25% | 28.6 | 19.1 | 14.9 |
| | 50% | 19.8 | 20.5 | 11.2 |
| | 75% | 24.4 | 20.0 | 11.9 |
| | 100% | 20.1 | 22.9 | 15.5 |
| high | 0% | 28.7 | 14.1 | 10.7 |
| | 25% | 24.6 | 10.1 | 6.4 |
| | 50% | 10.7 | 19.1 | 6.0 |
| | 75% | 20.3 | 18.9 | 5.54 |
| | 100% | 3.2 | 25.4 | 5.2 |

Table 12: DrQ+BC can adapt to multiple different distractions but is extremely brittle to settings it has not seen and struggles to generalize. Final mean performance is averaged over six random seeds and base undistracted performance is highlighted. The environment used is cheetah-run medexp.

| Shift Severity | % Shifted | Evaluation Return | | |
|---|---|---|---|---|
| | | Original | Distraction Train | Distraction Test |
| low | 0% | 79.1 | 0.5 | 1.1 |
| | 25% | 73.9 | 77.5 | 8.7 |
| | 50% | 72.3 | 82.2 | 8.5 |
| | 75% | 67.6 | 85.5 | 7.6 |
| | 100% | 4.2 | 86.6 | 4.5 |
| moderate | 0% | 79.1 | 0.2 | 0.9 |
| | 25% | 74.6 | 77.9 | 3.1 |
| | 50% | 68.0 | 83.6 | 3.4 |
| | 75% | 55.4 | 86.1 | 3.7 |
| | 100% | 0.9 | 86.7 | 2.0 |
| high | 0% | 79.1 | 0.1 | 0.5 |
| | 25% | 76.7 | 76.8 | 1.3 |
| | 50% | 66.7 | 82.5 | 1.1 |
| | 75% | 48.4 | 84.8 | 1.2 |
| | 100% | 0.4 | 86.1 | 0.9 |

### D.2 Further Offline DV2 Results on Multitask Datasets

To complement the results in Table 3, we show additional results for Offline DV2 on medexp multitask data. Here, Offline DV2 learns a slightly reduced quality policy compared to that on the base environment, but experiences no deterioration in performance on the test environments in walker or cheetah.

Table 13: Evaluation on the DMControl-Multitask benchmark using *medexp* data for Offline DV2. Normalized performance from [0, 1000] to [0, 100] is averaged over six seeds. Offline DV2 shows a strong ability to generalize to the extrapolation test environments.

| Algorithm | Environment | Eval. Return | | |
| --- | --- | --- | --- | --- |
| | | Train Tasks | Test Interp. | Test Extrap. |
| Offline DV2 | walker | 23.2 | 16.5 | 19.8 |
| | cheetah | 8.2 | 7.2 | 9.6 |

### D.3 DrQ+BC Random-Expert Ablation Studies

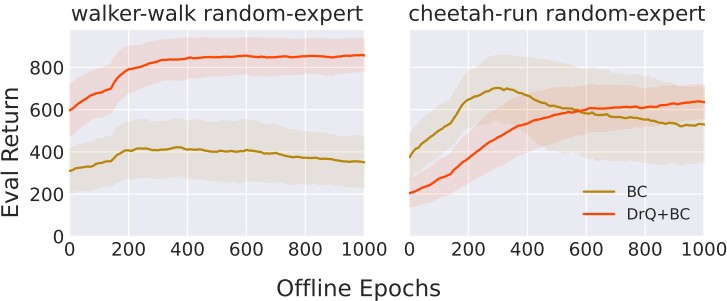

Figure 6: Comparison of DrQ+BC and BC on random-expert datasets with 500,000 of each datatype, results averaged over six seeds.

Analogously to Fujimoto & Gu (2021), we continue to see that DrQ+BC does well on mixed datasets with high reward with experiments on the concatenated random-expert datasets. Surprisingly, we see behavioral cloning also does reasonably well on cheetah-run with random-expert data. This is likely to be because there is significant distribution shift between the states of random and expert trajectories for that environment. This would lead to minimal destructive interference between similar states which contain completely different actions, as these state distributions largely do not overlap. This is in contrast to the medium-expert dataset, which experiences higher state overlap between its two modes (i.e., states generated by a medium and an expert policy respectively), resulting in a marginal "average" action being learned which is likely suboptimal, as can be seen by the poor performance of the BC agent in Tables 1 and 4.

### D.4 Analysis on World Models for Visual Observations

In this section, we seek to better understand the differences between model-based and model-free algorithms by presenting further analysis for the Offline DreamerV2 algorithm. In particular, we begin to address our open questions and understand why Offline DreamerV2 deals well with unseen distractions but has trouble handling larger or more narrowly concentrated datasets.

#### D.4.1 Model Training Time

One of the major factors preventing algorithms that use an RSSM like Offline DV2 and LOMPO from scaling to larger datasets is the time required for model training. The standard number of epochs of model training for our 100,000 datasets in Section 4 is 800 epochs, which takes around six hours on a V100 GPU. This scales

linearly with the number of training points if we maintain the same batch size. As we show in Figure 7, this is mandatory for performance and is a fundamental limitation of current model-based methods. We can see that the evaluated return increases and becomes more tightly distributed as the number of training epochs increases up to 800.

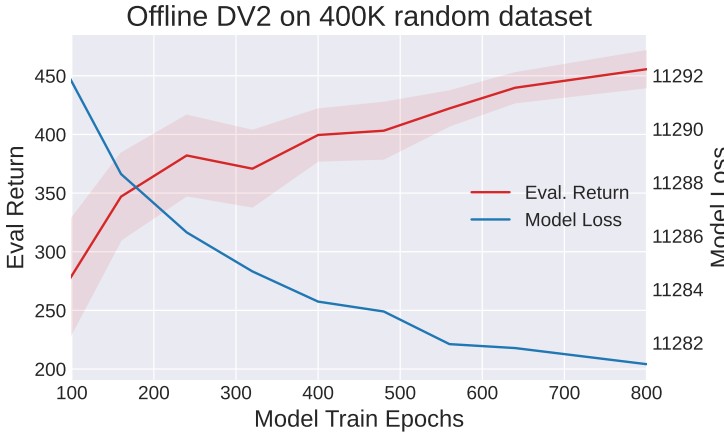

Figure 7: Evaluated return against model train epochs for Offline DV2 on a random dataset of size 400,000. We train a single model up to 800 epochs and evaluate the model periodically on three random seeds. We see that full model training, which scales linearly with training points, is mandatory for good performance.

### D.4.2 Uncertainty Quantification

In Table 1, we see that Offline DV2 is considerably weaker on the expert datasets that have narrow data distributions. On these expert datasets, we found that the model uncertainty often had lower variance across a trajectory compared to models trained on other datasets, and was thus uninformative. We give summary statistics for the uncertainty penalty on states generated by a random policy in the walker-walk environment in Table 14. Since we followed the author's DreamerV2 implementation (Hafner et al., 2020b), only the stochastic portion of the latent state is predicted by the ensemble; this may mean crucial calibration is lost when ignoring the impact of the deterministic latent. Future work could involve investigating SVSG (Jain et al., 2021), an extension to DreamerV2 with a purely stochastic latent state.

Table 14: Mean and standard deviation of the uncertainty penalty computed over 1,024 states sampled from the 'random' dataset on the walker-walk environment. Note that the model trained on expert data reports considerably smaller and tighter uncertainty values (compared to 'medium' and 'medexp'), despite the large distribution shift that exists from 'random' to 'expert' data. Instead, we'd expect the 'expert' trained model to exhibit the largest mean uncertainty when tested on the 'random' data.

| Dataset Type | Mean | Std. |
|---|---|---|
| random | 0.223 | 0.040 |
| mixed | 0.226 | 0.035 |
| medium | 0.341 | 0.034 |
| medexp | 0.338 | 0.034 |
| expert | 0.262 | 0.020 |

### D.5 Understanding Model-Based Extrapolation

Finally, we investigate the reasons why our model-based baseline, Offline DV2, appears to generalize to unseen distractions as seen in Section 5.1. The RSSM includes a latent decoder for the visual observations, which is primarily used during training to provide a self-supervised reconstruction loss. During deployment, the

policy solely relies on the latent states from the encoder, and the decoder is effectively discarded. However, we can still use the decoder at test time in order to understand the information contained in the latent states. To do this, we take the latent states generated during rollouts in the extrapolation experiments, and 'translate' them into natural images by passing them through the decoder.

We first investigate the setting where each RSSM is trained only on data from a single distraction and then transferred. This setting is the most successful, as can be seen in Figure 3. Thus, in Figure 8 we first show the ground truth observation provided to the agent, then below this we show the decoder output reconstructed from the latent. In many cases, we can recover the original pose despite ending up in different visual surroundings. This is an indication that despite the decoder overfitting to the exogenous factors in the visual input (e.g., background, colors), the latent captures the salient *state* information of the agent (e.g., joint positions and angles), explaining the strong test-time transfer performance.

In Figure 3, we note that settings with a mixture of distractors often had worse test-time transfer performance than those with a single distractor. To explain this, we consider an RSSM trained on images with a combination of two fixed distractors, and examine the reconstruction of an episode under a third distraction. We see in Figure 9, that the RSSM latents are split between the two modes of the data and the reconstruction switches robot color and background midway. This significantly confuses the recovered pose of the walker and likely causes a degradation in performance. Disentangling the factors of variation (Burgess et al., 2018; Mathieu et al., 2019) represents important future work for offline RL from visual observations.

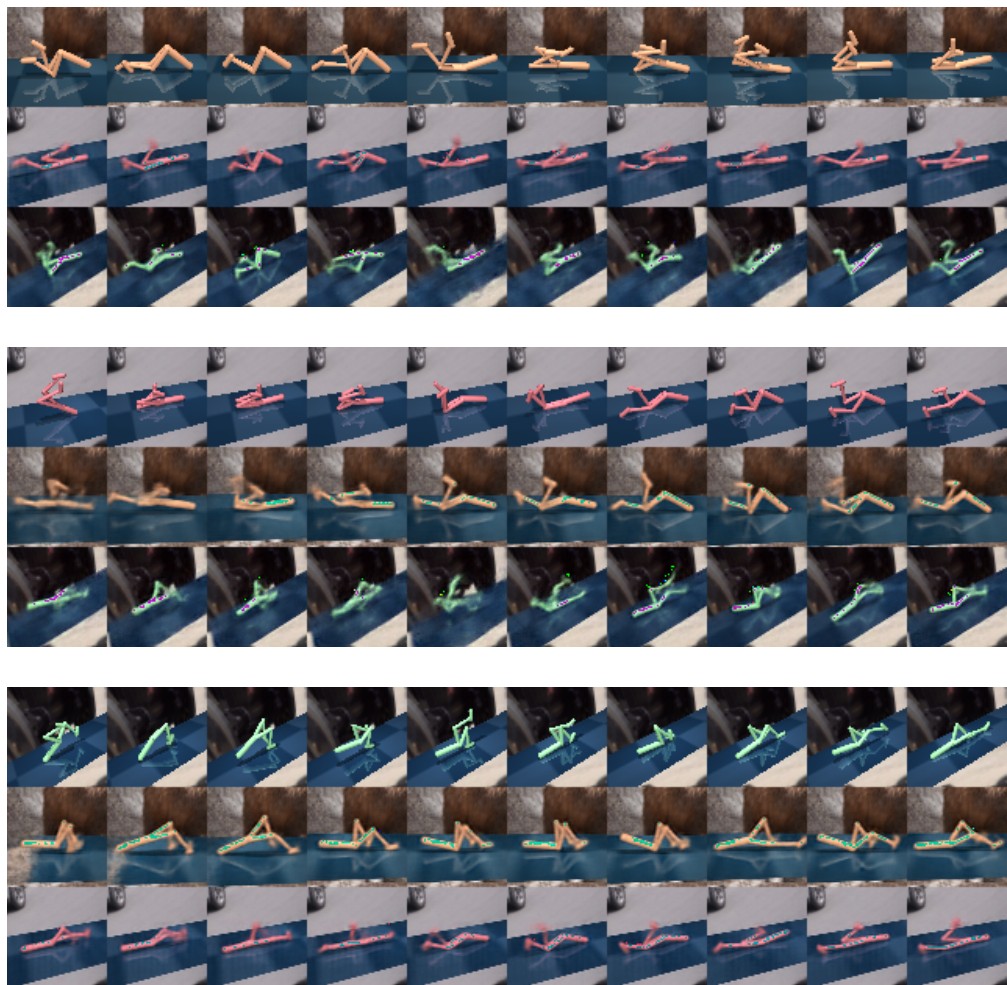

Figure 8: Reconstruction of random episodes shifted by a fixed distractor by models trained on data shifted by a different distractor. The top row shows the original ground truth, and the bottom two rows the model reconstruction for a different RSSM. We can see that in many cases, the original pose of the walker is still able to be recovered.

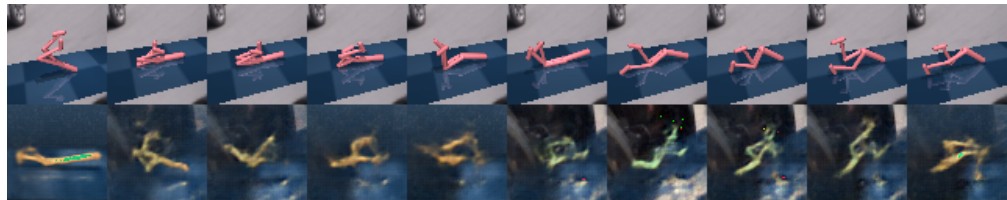

Figure 9: Bottom row shows the reconstruction of a distracted random episode (on the top row) using an RSSM trained on a mixture of two differently distracted environments. We can see that the reconstructed states are split between the two modes of the data and the reconstruction switches robot color and background midway.

