# OpenReview forum: "Challenges and Opportunities in Offline Reinforcement Learning from Visual Observations"
_TMLR — Accepted by TMLR_

### Review · Reviewer_W9te · 2023-04-25

**Summary Of Contributions:**

This paper presents an offline RL benchmark with visual observations by considering some key features/requirements that should be satisfied. Two baselines evaluation on the presented benchmark is also provided.

**Audience:**

Yes

**Broader Impact Concerns:**

Not applicable.

**Claims And Evidence:**

Yes

**Requested Changes:**

Details about the dataset should be clarified.

More in-depth insights should be provided.

Some statements should be carefully clarified, in more appropriate language.

More details are included in the pros and cons part.

**Strengths And Weaknesses:**

Strengths:

A thorough analysis of what characteristics should have for an offline RL benchmark is provided, together with empirical evaluations.

This paper proposes some simple model-based and model-free baselines for offline RL from visual observations, which are based on the classic algorithms DreamerV2 and DrQ under online learning. The authors have made reasonable and effective optimizations for offline training. The experimental results show the advantages of the model-based algorithm in datasets with broad data distributions, such as random/mixed, and the advantages of behavior cloning in datasets with high data quality, which are consistent with expectations.

Weaknesses:

Despite the detailed introduction to the dataset and baseline algorithms in this paper, there are still some areas that need improvement to enable researchers to efficiently conduct further studies based on this paper, from a more detailed perspective. The main shortcomings of the paper are as follows:

1. Regarding the dataset, some details are not clearly introduced, which may cause confusion for researchers.

>a) It is recommended to add a column to Table 1 to show the mean return of the dataset or provide results under an expert policy similar to D4RL.

>b) Sections 5.1 and 5.3 introduce new datasets (datasets with Visual Distractions and larger-scale datasets) for evaluation, but the paper does not provide information about these datasets. Appendix A only provides information about the 15 basic datasets. Therefore, it would be better to list all available datasets and their characteristics.

2. Open Problem 3 is not a problem unique to visual observation offline RL, but rather a problem of extrapolation error caused by unseen data, which is common in offline RL, and more specifically, offline model-based RL. The author needs to provide a more in-depth discussion and insight.

3. Other details:
3a. DreamerV2 and DrQv2 are no longer state-of-the-art vision-based online reinforcement learning algorithms, and this statement should be corrected.

---

> ### Author Response · Authors · 2023-06-03
> **Response to W9te**
>
> Thank you for your detailed review. We were pleased to see you found our analysis thorough! We hope this response addresses the concerns and questions raised in your review. Please let us know if any other issues remain.
>
> **“Adding a column to table 1 to show the mean return of the dataset”:**
>
> We previously included the mean return of each dataset in Table 5 in the Appendix. However, we are also happy to add it to Table 1 for further clarity. Thank you for the suggestion.
>
> **More information on datasets with Visual Distractions and larger-scale datasets**
>
> Thank you for raising this point, we explain in Section 5.1 that the statistics on the visually distracted datasets are exactly identical to the original datasets. Thus, we did not feel it necessary to duplicate this information. Similarly, since the larger-scale datasets are generated using the same behavioral policy but just more samples from the same distribution, the statistics of the dataset are next to identical to the original.
>
> **Open Problem 3 is not a problem unique to visual observation offline RL**
>
> The reviewer is right that generalization to unseen data is a common problem in offline RL and this is even true more broadly across all areas of machine learning. However, we thought the statement was particularly appropriate to conclude Section 5.2 as the analysis in the section showed two interesting questions specific to the algorithms and baselines we introduced. First, the difference between the generalization capability between our model-free and model-based baselines. Second, the differences in OOD and in-distribution generalization for our DrQ+BC algorithm. As we highlighted in the paper, we believe both of these represent interesting future research questions.
>
> **DreamerV2 and DrQv2 are no longer state-of-the-art vision-based online reinforcement learning algorithms**
>
> Thank you for raising this, we have adjusted the statement to be popular in the abstract and the introduction.

---

### Review · Reviewer_jAZh · 2023-05-31

**Summary Of Contributions:**

This paper introduces a novel benchmark called "v-d4rl: Vision Datasets for Deep Data-Driven RL" which focuses on offline RL using visual observations of DMControl Suite (DMC) tasks (Tassa et al., 2020). This benchmark builds upon the design principles of the popular d4rl benchmark (Fu et al., 2021). It is the first publicly available benchmark for continuous control that encompasses a wide range of behavioral policies.

Second, to address the challenges of realistic offline RL from visual observations, this paper identifies three crucial aspects: robustness to distractions (Stone et al., 2021), generalization across dynamics (Zhang et al., 2021), and improved performance for offline reinforcement learning at scale. To establish baselines for offline RL from visual observations, this paper adapted state-of-the-art online RL algorithms, namely DreamerV2 (Hafner et al., 2020b) and DrQ-v2 (Yarats et al., 2021a). These modified algorithms serve as simple baselines for measuring progress and guiding future advancements in this domain.






**Audience:**

Yes

**Broader Impact Concerns:**

I do not observe any explicit discussion on the Broader Impact Concerns of the topic studied.

**Claims And Evidence:**

Yes

**Requested Changes:**

Please address accordingly as suggested above.

**Strengths And Weaknesses:**

Strengths: This paper creates a new dataset that fulfill the existing gap of missing public availabile visual observations for offline reinforcement learning. In this sense, this paper is very intriguing. It systemically considers both model-based and model-free algorithms, which provides a complete study for RL. It studies several key aspects including: Robustness to Visual Distractions, Generalization Across Environment Dynamics, and Improved Performance with Scale.

Weakness: although the v-d4rl dataset is a nice contribution, the environment included is limited. Concretely, only walker-walk, cheetah-run, and humanoid-walk are included, which makes the scale of the dataset small. From my opinion, more experiments/environments are needed for this work to receive higher attention.

---

> ### Author Response · Authors · 2023-06-03
> **Response to jAZh**
>
> Thank you for your thoughtful review. We were pleased that you believe that our dataset fulfills the existing gap of missing public benchmarks in this field. We hope this response addresses the concerns and questions raised in your review. Please let us know if any other issues remain.
>
> **Scale of V-D4RL**
>
> Thank you for raising this point. The main evaluation of V-D4RL covers 15 datasets across 3 environments, which we note is the same number as the main MuJoCo D4RL environments (also 15 datasets across 3 environments) which has driven years of development and research in offline RL from proprioceptive states [1, 2, 3]. Thus, we believe that rather than being restrictive, V-D4RL has the potential to be an equally approachable driver of progress in this field. Furthermore, the inclusion of the visually distracted and multitask environments more than triples the number of base datasets for more specialized evaluations.
>
> [1] MOPO: Model-based Offline Policy Optimization. Tianhe Yu, Garrett Thomas, Lantao Yu, Stefano Ermon, James Zou, Sergey Levine, Chelsea Finn, Tengyu Ma.
>
> [2] Uncertainty-Based Offline Reinforcement Learning with Diversified Q-Ensemble. Gaon An, Seungyong Moon, Jang-Hyun Kim, Hyun Oh Song.
>
> [3] Offline reinforcement learning with fisher divergence critic regularization. Ilya Kostrikov, Rob Fergus, Jonathan Tompson, Ofir Nachum.

---

### Review · Reviewer_YCLk · 2023-06-01

**Summary Of Contributions:**

This paper presents a standardized benchmark and baselines
for offline reinforcement from visual observations using
the walker-walk, cheetah-run, and humanoid-walk environments
from the DeepMind control suite.
Section 2 describes the basic offline RL setup.
Section 4.1 describes the environments, how expert
policies were obtained by training a SAC agent on the
proprioceptive states, and how the data was obtained from
the expert's replay buffer.
Section 4.2 describes a model-based baseline based on DreamerV2
and a model-free baseline based on DrQ-v2 with a behavioral
cloning regularization.

**Audience:**

Yes

**Broader Impact Concerns:**

I do not have any concerns

**Claims And Evidence:**

No

**Requested Changes:**

**RC1.** Clarify the inconsistent and incomplete comparisons to LOMPO (for W1/W2)

**RC2.** Normalize the scores using the dataset (as in D4RL) or make
   the dataset's mean reward more visible, e.g., as a reference line in Figure 2 (for W3)

I have one other question out of curiosity:

**Q1.** Did you consider adding BC to the policy in Dreamer-v2?


**Strengths And Weaknesses:**

# Strengths

Standardizing benchmarks for visual offline RL is an
important topic as papers are otherwise forced to use
their own datasets or datasets privately sent from the
authors of other papers. Creating this benchmark is a
serious undertaking that I commend the authors for.
I believe the paper has a great potential to be a snapshot
of the current state of methods and results in this space.

# Weaknesses

The main weakness is in the comparison to LOMPO:

**W1.** The numbers in Table 2 should correspond to
   [Table 1 of the LOMPO paper](http://proceedings.mlr.press/v144/rafailov21a/rafailov21a.pdf),
   but the LOMPO results in this paper do not match the originally
   published results. What is causing this difference?
   This makes the claim of Offline DV2 and DrQ+BC outperforming LOMPO
   potentially not sound because the originally published LOMPO
   results give better performance in some settings.

**W2.** The walker-walk is only one setting considered in LOMPO:
   Omitting the other environments used in the LOMPO paper
   also make the claim of Offline DV2 and DrQ+BC outperforming LOMPO
   potentially not sound.

One other minor weakness is:

**W3.** It seems slightly confusing to map from the original environmental
   rewards from [0, 1000] to [0,100]. I much prefer the normalized
   scores, e.g. from D4RL, that normalize using the dataset's score.
   It would be nice to add the expert's performance to the figures
   in some way because there is a significant gap:
   Table 5 shows the expert on humanoid-walk attains a
   mean reward of 858 while Figure 2 shows that the best model
   isn't able to surpass 100.

And an extremely minor weakness:

**W4.** I do not find the open problem statements very interesting as
   they are generically applicable to most
   machine/reinforcement learning settings
   (1. can a single algorithm outperform all baselines? and
   2/3. how to improve generalization to visual
   distractions/unseen dynamics?)

---

> ### Author Response · Authors · 2023-06-03
> **Response to YCLk**
>
> Thank you for thorough and insightful review. We were pleased that you appreciated the importance of the benchmark and the potential of the paper to serve as a snapshot of the current state-of-the-art. We hope this response addresses the concerns and questions raised in your review. Please let us know if any other issues remain.
>
> **Comparison to LOMPO**
>
> Thank you for raising these two points. These points were some of the main motivations for our creation of a fully open-source, open data, and transparent benchmark. First, we address the numbers in Table 2 not corresponding to Table 1 of the LOMPO paper. The method for normalizing scores in the LOMPO paper is never fully disclosed as their reference scores were never made public as they were for D4RL. Thus, we were unable to interpret the original LOMPO numbers in a meaningful way. Therefore, we had to rerun the experiments under our normalization scheme.
>
> Second, on the omission of other environments used in LOMPO. The datasets used in the LOMPO evaluation were never publicly released and we could only evaluate on what data we received from the authors of the original paper. This is precisely the gap in the field that our paper was designed to fill.
>
> **Making the dataset's mean reward more clear**
>
> Thank you for this suggestion - we have now done this in Table 1 where we present the full results of our main evaluation.
>
> **Did you consider adding BC to the policy in Dreamer-v2?**
>
> Thank you for the suggestion, we did not but believe this could be a very promising approach to combining the strengths of model-free and model-based methods for future work!

---

> > ### Comment · Reviewer_YCLk · 2023-06-03
> > **Response**
> >
> > Thank for you the prompt response and updates. Did you ask the LOMPO authors for how they normalized their experimental results or to send you the remaining datasets? It would be extremely interesting and useful to the community to have a direct comparison to the results they report, and to better-understand or confirm any discrepancies.

---

> > > ### Author Response · Authors · 2023-06-03
> > > **Response**
> > >
> > > We did, and we did not receive any further response from the LOMPO authors on those questions. Thus, we believe the reproduction of the LOMPO results on DMC walker-walk with their code and their data over 6 seeds to be the most accurate reference we have. We apologize if this is a slightly unsatisfying response to your question.
> > >
> > > We hope that our benchmark provides a way forward for the community to have fully reproducible and reliable results for benchmarking progress in offline RL from visual observations.

---

> > > > ### Comment · Reviewer_YCLk · 2023-06-03
> > > > **Response**
> > > >
> > > > Ok, thank you for the clarifications! I have no other outstanding concerns with the paper and still think it's a good direction to help standardize methodologies. I encourage the authors to try finding out how LOMPO normalized their rewards to enable a direct comparison between the results

---

> > > > > ### Author Response · Authors · 2023-06-10
> > > > > **Thanks!**
> > > > >
> > > > > Thanks for the responses and we’re glad we could clarify your concerns. If we hear back from the LOMPO authors on how they normalized returns, we are more than happy to amend our paper.

---

### Author Response · Authors · 2023-07-13
**Camera-ready version**

We would like to thank the reviewers and action editor for valuable feedback during the review process which greatly improved our paper. We have now uploaded the camera-ready version incorporating all the feedback to date as well as a link to the codebase.

---

### Decision · Action_Editors · 2023-07-03

**Recommendation:** Accept as is

**Comment:**

All reviewers recommend acceptance and the AE agrees.

**Audience:**

There is a clear audience for this work in the TMLR community.

**Claims And Evidence:**

After some discussion with the authors, all reviewers agree the claims are supported by appropriate evidence.